# Lateral Capacitance–Voltage Method of NanoMOSFET for Detecting the Hot Carrier Injection

**Atabek E. Atamuratov [1],*, Ahmed Yusupov [2], Zukhra A. Atamuratova [1],
Jean Chamberlain Chedjou [3] and Kyandoghere Kyamakya [3]**

[1]  Physics Department, Urgench State University, Kh. Olimjan Str., 14, Urgench 220100, Uzbekistan;
atamuratovazuhra03@gmail.com
[2]  Department of Electronics and Radio Engineering, Tashkent University of Information Technologies,
A. Temur Str., 108, Tashkent 100200, Uzbekistan; a.yusupov@tuit.uz
[3]  Department of Transportation Informatics, University of Klagenfurt, 9020 Klagenfurt, Austria;
jean.chedjou@aau.at (J.C.C.); kyandoghere.kyamakya@aau.at (K.K.)
*  Correspondence: atabek.atamuratov@yahoo.com; Tel.: +998-999-631-863

**Abstract:** In this paper, the dependence of the capacitance of lateral drain–substrate and source–substrate junctions on the linear size of the oxide trapped charge in MOSFET is simulated. It is shown that, at some range of linear sizes of the trapped charge, the capacitance of lateral junctions linearly depends on the linear size of the trapped charge. The dependence of the difference between drain–substrate and source–substrate capacitances on the linear size of trapped charges is also simulated. The revealed dependence can be used in measurements to estimate the linear size of oxide trapped charges induced by hot carrier injection, which can occur during MOSFET operation at defined conditions.

**Keywords:** MOSFET; degradation; lateral transition capacitance; threshold voltage; local charge

## 1. Introduction

During operation, a MOSFET is exposed to different types of electrical stresses, which can lead to a degradation of its characteristics. Among these are noticeably some inherent and well-known aging effects related to hot carrier injection (HCI) [1,2], bias temperature instability (BTI) [3,4], and OFF-state stress [5,6], as well as the transient effects of random telegraph noise (RTN) [7,8], which must be considered (or taken into account) into the design of digital and analog integrated circuits. In principle, such degradation effects arise through the injection of charge carriers into the oxide or through trapping at the oxide–semiconductor interface. In an n-MOSFET biased by a low gate voltage and a relatively high drain voltage (Vg << Vd), the rate of hot electron injection into the gate oxide near the drain from the channel is high [9–12]. As a result, the charge is mainly collected in the gate oxide layer or at the interface near the drain. However, some hot holes can also be injected into the oxide [13–17].

In typical analog applications, however, the transistor is likely to operate at gate voltages just above the threshold (Vg ≈ VT). Accurate measurements of the induced gate currents and simulations of the device show that the field distribution at low gate voltages enhances the injection of a small part of hot holes into the gate oxide, which, after trapping, act as fixed positive charges in/of the oxide [18–20]. In this case, the charge is trapped in a larger area [21].

The charge trapping in the oxide or at interface leads to a shift in device parameters such as threshold voltage, transconductance, and subthreshold slope. Consequently, the device performances as well as circuit performances are significantly affected [22]. Therefore, it is very important to develop new methods to define the presence and distribution, along the channel oxide and the interface,

of charges induced by electrical stress. Simplicity, flexibility, and speed of the method are also important, as a large-scale aging test must be performed on hundreds of measured transistors in order to obtain sufficient statistics. There are several methods to extract the distribution of traps; however, each has specific drawbacks. The most popular among them is the charge pumping method, but it can only be used when the substrate current is measurable [23,24]. Further, a method that compares drain current in linear and saturation regions of operation gives only the relative location (i.e., a very rough distribution) of traps [25,26]. A method to estimate the profile of the threshold voltage change due to electrical stress requires only Id-Vg measurements, but DIBL and a charge-sharing effect should also be considered to improve accuracy [27].

This work discusses the physical basics of a very simple and fast method to diagnose oxide and charge trapped at the interface in a nano-MOSFET by using the simulation results. The experimental background for the method is provided by our previous works [28,29]. These works show that the uneven (non-uniform) charge distribution in the oxide layer and at the oxide–semiconductor interface of the MOSFET leads to a change in the capacitances of the lateral drain–substrate and source–substrate junctions. Therefore, the distribution of charges along the canal significantly affects the values of the aforementioned capacitances.

This paper is organized as follows. In Section 2, we present the description and details of the capacitances of source–substrate and drain–substrate lateral junctions. Simulation methodology and simulation results with discussions are described in Sections 3 and 4, respectively. Finally, conclusions are summarized in Section 5.

## 2. Capacitances of Lateral Junctions

The proposed method is based on the measurement of capacitance dependence on the voltage of the drain–substrate and source–substrate lateral junctions. The junction capacitances can be split into a side wall portion $C_{sw}$ and a bottom wall portion $C_{bw}$:

$$C_{ds} = C_{sw} + C_{bw} \tag{1}$$

The same expression (see Equation (1)) can also be written for $C_{ss}$. The capacitance associated with the side wall portion is obtained by multiplying the length of the side wall perimeter by the effective side wall capacitance per unit length. The capacitance for the bottom wall portion is obtained by multiplying the area of the bottom wall by the bottom wall capacitance per unit area. The side wall portion of the capacitance $C_{sw}$, in turn, consists of two parts, namely the capacitance associated with the side wall of the contactless channel $C_{swn}$ and the capacitance associated with the side in contact with the channel $C_{swc}$:

$$C_{sw} = C_{swn} + C_{swc} \tag{2}$$

Note that all parts of the lateral junction capacitance $C_{ds}$, as well as $C_{ss}$, depend on the geometry of the junctions and the applied voltage. Besides, the part of a side wall capacitance connected with the side in contact with the channel $C_{swc}$ depends on the potential profile of the channel, which is controlled by the applied voltage and is influenced by the charge trapped in the oxide layer or at the interface. Therefore, the potential profile depends on the linear size of the trapped charge domains along the channel. Small signal drain–substrate capacitance ($C_{ds}$) and source–substrate capacitance ($C_{ss}$) are functions of the transistor's channel potential profile, which is controlled by the applied bias. The presence of trapped charges in the oxide layer or at the interface leads to a local change of this potential and consequently to capacitances $C_{ds}$ and $C_{ss}$, which are different from those without a trapped charge. Measurement of changing the capacitance before and after charge trapping is therefore a practical tool for studying the properties of the trapped charge due to hot carrier injection. Therefore, the main objective of this work is to simulate the capacitances of lateral junctions $C_{ds}$ and $C_{ss}$, analyze the dependences between these capacitances, and analyze the linear size of the charge-trapped

domains in the oxide layer at the drain end. The dependence studied can be used in measurements to estimate the linear size of the trapped charge induced by electrical stress.

## 3. Simulation Methodology

The effects of electrical stresses depend on the value and polarity of the applied voltage and the duration of the stress time. These effects result in trapping of the charge in the oxide layer or/and at the interface. The expansion of this charge occurs mainly along the channel, while along the width of the channel the trapped charge is distributed homogeneously. Therefore, the effects of electrical stress are primarily a 2D problem rather than a 3D problem. In 2D simulation, the width of the device suitable for the third dimension in commercial software tools/programs, as well as in the TCAD Sentaurus used in this work, is usually 1 μm. Further, in many papers that take into account the effects of electrical stresses, the different values of the widths are in a range that includes the value 1 μm (e.g., 0.47 μm and 0.87 μm in [27], 0.81 μm in [25], 10 μm in [30]). Let us mention, however, that the dependence on the width is beyond the scope of this work.

In addition, the 2D simulation takes less time. This is why we favored 2D simulation in our research. Two-dimensional simulations were carried out based on the Advanced TCAD Sentaurus program package. Two simulation methods were used in this work. The drift–diffusion model with allowance for the dependence of mobility on the doping, the saturation of the carrier velocity at high fields, and the influence of normal field component on the drain current was used for simulating threshold voltage dependence on a trapped charge area. The small signal AC analysis method was used for simulating C–V dependences of the side source–substrate and drain–substrate junctions. The frequency of the AC signal was selected as 1 MHz.

The gate length of the simulated MOSFET 100 nm and the gate oxide thickness 3.7 nm were chosen from the range in which the main electrical stresses have so far been considered in the literature [25,27,30]. The doping level of the p-Si substrate (base) was $5 \times 10^{15}$ cm$^{-3}$. Our analysis consists of embedding a local positive charge with different linear sizes (d) along the channel into the gate oxide and defining its influence on the capacitance of the lateral drain–substrate and source–substrate junctions. At hot carrier injection (hypothesis at stake in this work), the oxide-trapped charge is concentrated at the drain end and is afterwards extended in the direction to the source region as the stress time increases. Therefore, the linear size of the charged area corresponds to the length of the charge distribution from the drain region along the channel towards the source region. The charge trapped in the gate oxide layer was modeled by a homogeneously charged area. In fact, the density of oxide-trapped charge induced by electrical stress has some distribution along the channel, depending on the stress condition [30,31]. As our main objective is to check a law expressing the dependence of the capacitance of the lateral junctions on the linear size of the trapped charge, we considered (as assumption) the area charged homogeneously with an average density of trapped charge. Indirectly, to estimate how this assumption influences the results, we considered the dependence of the change in threshold voltage on the linear size of the homogeneously charged area (see Section 3). The results obtained were in good qualitative agreement with the experimental results in [30]. An evaluation/estimation and a quantitative comparison of the capacitance changes obtained from simulations and experiments were carried out. It can be seen (through simulation) that the maximum relative changes in the capacitance of the lateral drain–substrate were about 6%. Under certain conditions for introducing a charge into the oxide, this value was in the range of variation of the lateral capacitance carried out in experiments (see Figure 2 in [28] and Figure 1 in [29]).

The density of oxide traps in the charged area was chosen to be $10^{12}$ cm$^{-2}$, which can take place at hot carrier injection stress in MOSFETs [28]. The changes in threshold voltage $\Delta V_{th}$ and capacitances of lateral depletion regions $\Delta C_{ds}$ and $\Delta C_{ss}$ were simulated for different sizes of the local oxide charge.

## 4. Simulations Results and Discussions

The simulation result reveals that embedding a positive charge led to decreasing the threshold voltage, which depended on the size of the local charge (Figure 1). This effect is one of the main indicators of electrical stress in MOSFETs. Mainly in the experiments, the dependence of the change in threshold voltage on the stress time was measured. However, it is well known that increasing the stress time, under defined conditions, leads to an expansion of the charged area. In order to verify the conformity of the physical models used in the simulation with real cases, the dependence ΔVth on the length of the oxide-trapped charge was simulated. In the considered range (see Figure 1), the decrease of the threshold voltage with an increase in the linear size (d) of the oxide-charged area was in qualitative agreement with the experimental results in [28]. Here, the dependence of the threshold voltage on the stress time is investigated.

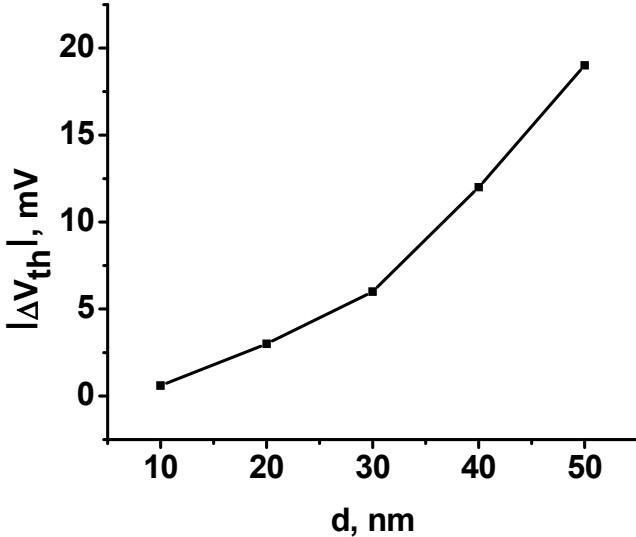

**Figure 1.** Dependence of the threshold voltage change on the linear size of charged area.

As expected, embedding a charge into the gate oxide layer led to a change in the effective capacitance of the drain–substrate and source–substrate lateral junctions in the corresponding C–V dependencies of these junctions at low voltages. The effective capacitance of the aforementioned junctions means that the C–V dependence of the lateral junctions is not similar to the C–V dependence of a separate p–n junction (Figure 2). This is obviously the result of the significant impact of other MOSFET junctions at small sizes. For comparison, the C–V dependency of the drain–substrate junctions was simulated for different gate lengths (Figure 2). For a gate length of more than 300 nm, the C–V dependence of the drain–substrate junction was similar to the C–V dependency of a separate p–n junction. At shorter gate lengths, the capacitances of lateral junctions correspond to the effective capacitances. Therefore, in our case with a gate length of 100 nm, we have taken into account the effect of the local oxide charge on the effective lateral capacitance.

The embedded charge located at the end of the gate oxide near the drain caused an increase in the lateral capacitance in the C–V dependence at low voltages (Figure 3). The presence of a trapped charge in the oxide layer resulted in a redistribution of charge carriers and, therefore, a curvature of the border/edge of the depletion layer of the lateral/side junction near the surface of the substrate. The curvature caused a change in the junction capacitance. To explain this effect, we can use the flat capacitor approach for lateral/side junctions. As mentioned above (see Equations (1) and (2)), the capacitance of the lateral drain–substrate junction can be expressed as the sum of three components:

$$Cds = Cswn + Cswc + Cbw \qquad (3)$$

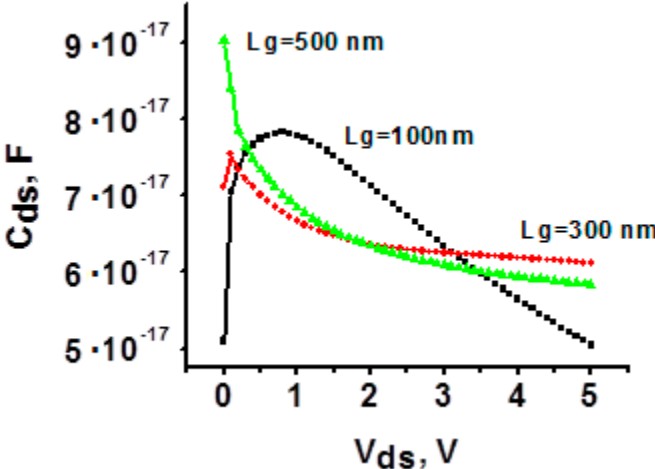

**Figure 2.** C–V dependences of drain–substrate junction at different gate lengths of MOSFET.

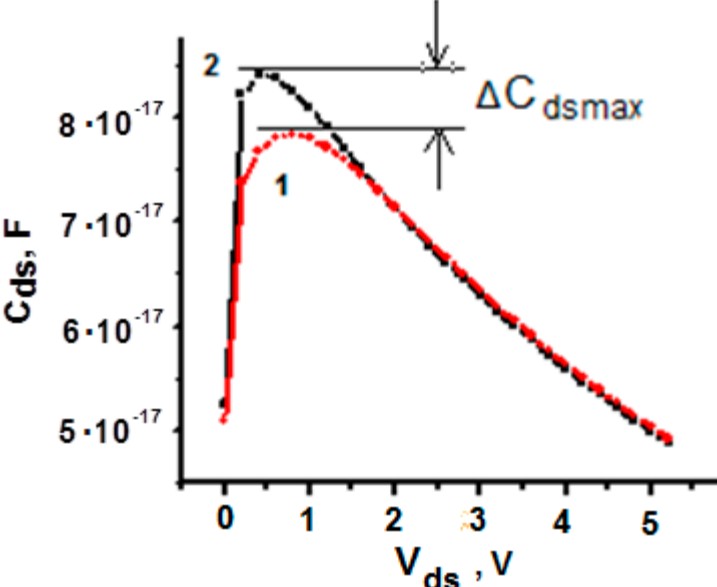

**Figure 3.** C–V dependences of drain–substrate junction without embedded charge (**1**) and at embedding the local oxide charge near the drain (**2**).

The same expression (see Equation (3)) can be written for the capacitance of the source–substrate junction. It is obvious that the trapped charge mainly influences the capacitance associated with the side wall of the junction in contact with the channel Cswc. The latter, in the flat capacitor approach, can be expressed by Cswc = $\varepsilon\varepsilon_0$S/W, where $\varepsilon\varepsilon_0$ is the dielectric constant of the substrate, and S and W are the area and the width of the depletion region of the junction, respectively. The curvature of the border of the depletion layer significantly affects the ratio S/W and therefore leads to changes in Cswc and Cds. At low voltages, the curvature of the considered border was considerable compared/relative to the border of other parts of the lateral junction (e.g., the part of the side wall not in contact with the channel and the bottom wall part) than at high voltages. This is clearly depicted in the C–V dependence in Figure 3 through a significant change of the capacitance at low voltages. Figure 4 shows a map of the distribution of minority charge carriers in the substrate for two cases: (a) the case considering the non-embedding of a local charge into the gate oxide and (b) the case considering the embedding of a local charge. This distribution is appropriate to the applied voltage $V_{ds} = 0.5$ V, which corresponds to the range of the maximum variation of the lateral capacitances in the C–V

dependence. The distribution patterns (in Figure 4) reveal that, in the case where a local oxide charge was embedded, the effective width of the depletion layer (the width of the transition region with an electron concentration exceeding $5 \times 10^{15}$ cm$^{-3}$) was narrower than in the case where no local oxide charge was embedded. Consequently, in case a local oxide charge was embedded, the capacitance was higher at the aforementioned specific value of the applied voltage ($V_{ds} = 0.5$ V). This is clearly depicted in the C–V dependence of the lateral transitions in Figure 3. At high applied voltages, the distribution of the charge carriers in the substrate did not practically differ in the cases mentioned above, as it is shown in Figure 3 (see the range of variation of the applied voltages in which the C–V dependency coincides in Figure 3).

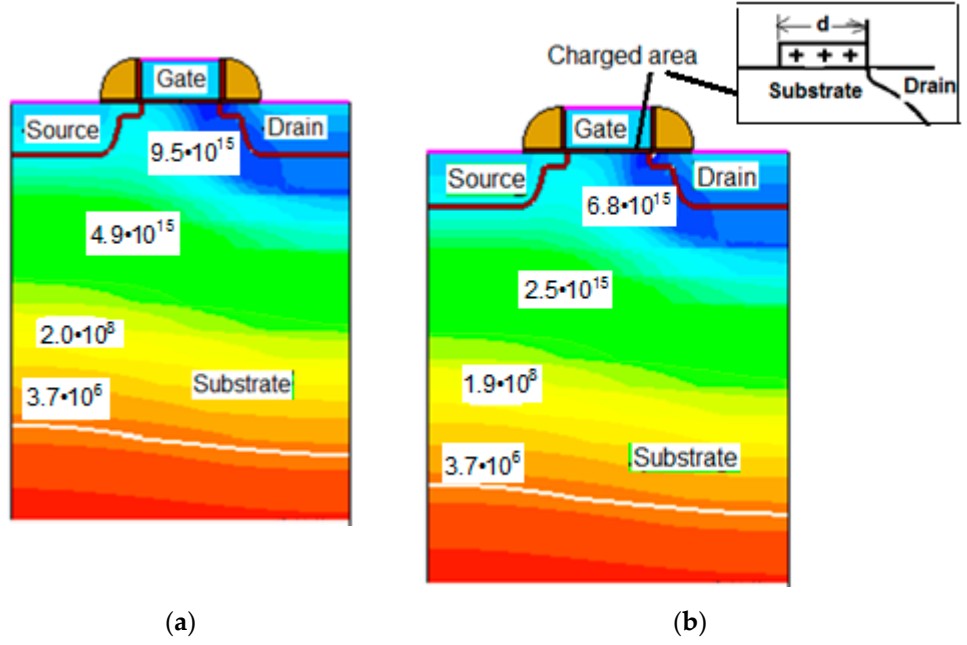

(**a**)　　　　　　　　　　　　　　　　　　　　　　　　　(**b**)

**Figure 4.** Minor carrier concentration distribution in substrate (base) in case of absence (**a**) and presence (**b**) of embedding local oxide charge. $V_{ds} = 0.5$ V and the indicated concentration is in cm$^{-3}$.

The amplitude of changes in capacitances $\Delta C_{dsmax}$ and $\Delta C_{ssmax}$ depends on the linear size (d) of the embedded local oxide charge (Figure 5). As can be seen from Figure 5, $\Delta C_{dsmax}$ increased linearly with increasing d. In contrast, after crossing the center/middle of the oxide along the channel, the capacitance $\Delta C_{dsmax}$ reached saturation (see top point in the red curve in Figure 5) and afterwards decreased with a further increase in d. This reveals the weakening of the influence of the local oxide charge on the redistribution of charge carriers in the vicinity of the drain when increasing the distance up to source. As it appears in Figure 5, the capacitance $\Delta C_{ssmax}$ also rose linearly with increasing d. This evolution reached saturation after passing through the middle of the oxide and kept increasing (at a significantly high rate) with a further increase in d. This reveals the increased effect of the local oxide charge on the redistribution of charge carriers in the vicinity of the source as the distance from the source decreases.

After reaching the end of the gate oxide, the local charge covered all the gate oxide, and consequently the capacitances of both side junctions were under the same conditions. Overall, Figure 5 shows the dependence of the amplitude of changes in capacitances ($\Delta C_{dsmax}$ and $\Delta C_{ssmax}$) on the linear size (d) of the embedded local oxide charge. As clearly depicted in Figure 5, the capacitances $\Delta C_{dsmax}$ and $\Delta C_{ssmax}$ at d = 100 nm were equal. Such behavior of the lateral capacitances eases the estimation of the linear size of the embedded oxide charge. The aforementioned dependence of the capacitances on the linear size had a peak (see the top point in Figure 5) that corresponds to the expansion of the charged area towards the middle of the oxide layer along the channel. This peak is a very important

critical point that can appropriately be used for the calibration of the dependence and to accurately estimate the size of the charged area. The linearity of the dependence in the first half of the entire range of d makes it easier to estimate the linear magnitude of the injected charge. Experimentally, by simply measuring the difference of the lateral capacitances, one can quickly estimate the dynamics of the injection of hot charge carriers into the oxide layer and/or the oxide–semiconductor interface that take place during operation of the MOSFET.

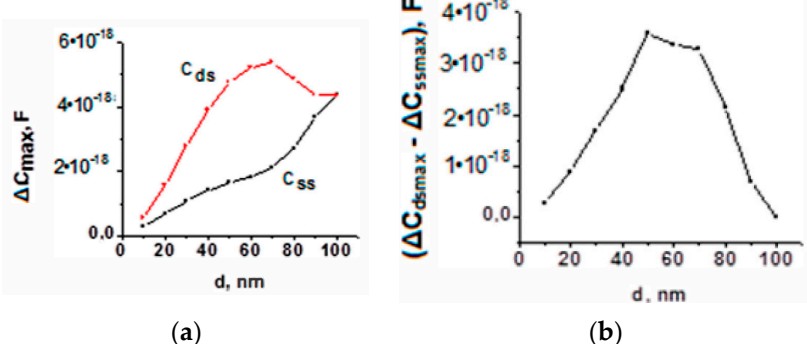

| (a) | (b) |

**Figure 5.** Dependence of the amplitude of changes in lateral capacitances ΔCdbmax and ΔCsbmax (**a**) and their difference (**b**) on linear size of the embedded charge at the drain end of the gate oxide.

## 5. Conclusions

This work has led to some interesting results that can be summarized as follows. It has been shown that the embedding of a local charge in the gate oxide in the vicinity of the drain causes a change in the capacitances of the drain–substrate and source–substrate lateral junctions of the MOSFET. The maximum amplitude of the changes in capacitance depends on the expansion of the charged area. The dependencies obtained can be used to estimate the linear size of the trapped oxide charge that can be caused by the injection of hot carriers during MOSFET operation. The simulation results (obtained in this work) could serve many purposes such as the characterization of various reliability issues related to MOSFETs degradation under electrical stress.

**Author Contributions:** Conceptualization, A.E.A. and A.Y.; methodology, A.E.A. and Z.A.A.; software, Z.A.A.; validation, A.E.A. and Z.A.A.; formal analysis, J.C.C.; investigation, Z.A.A.; resources, Z.A.A.; data curation, Z.A.A.; writing—original draft preparation, A.E.A.; writing—review and editing, J.C.C.; visualization, J.C.C., K.K. and A.Y.; supervision, K.K.; project administration A.E.A.; funding acquisition, A.E.A. All authors have read and agreed to the published version of the manuscript.

**Funding:** This research was funded by the Ministry of Innovative Development of the Republic of Uzbekistan, grant number OT-F2-67.

**Conflicts of Interest:** The authors declare no conflict of interest.

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
