# Peer review of "Lateral Capacitance–Voltage Method of NanoMOSFET for Detecting the Hot Carrier Injection"

_applsci, doi:10.3390/app10217935_

Round 1
Reviewer 1 Report
The abstract is too vague; please specify the main ideas in a more precise way.
The introduction is too short. There is a space to specify the actual state-of-the-art, to declare the authors’ way (differences, advantages or disadvantages compared to other authors’ work), etc.
Chapter 2 – Where are the (dis)advantages of the 2D model? How strong might be the effect of the third dimension (finite length of the structure)?
Compared to reality, there are simplifications. Can you please clarify them and evaluate their influence on results?
Generally, the article has several serious flaws. The background theory is missing in the article. The article is based on simulations only. No experiments are proofing the presented results (I would like to say “presented ideas”; nevertheless, they need to be set-up on theory, which is missing).
The word "measurement" is used at least three times in the article, including in the article title, which is confusing.
Technical/formatting comments: the quantities and units in graphs are not matching the technical standards. Quality of the graphs (Fig. 2 and 3) is relatively low. The legend is missing in some figures; resp., it is useless (e.g. “B” in Fig. 1).
Author Response
Many thanks to reviewer for the valuable remarks, notes and suggestions, which helped to draw attention and I hope to improve the manuscript.
The responses and comments to the reviewer's remarks:
1.The abstract is too vague; please specify the main ideas in a more precise way.
The abstract is changed totally with respect to the main ideas.
- The introduction is too short. There is a space to specify the actual state-of-the-art, to declare the authors’ way (differences, advantages or disadvantages compared to other authors’ work), etc.
The introduction is considerably extended by declaring the advantages of proposed method compared to other authors (line 44) and by describing the proposed method more precisely (lines 48 and 53)
3.Chapter 2 – Where are the (dis)advantages of the 2D model? How strong might be the effect of the third dimension (finite length of the structure)?
At line 55 of section 2 the possibility and advantages of the 2D model is explained:
Effects of electrical stresses depend on the value and polarity of the applied voltage and the duration of stress time. These effects results in trapping the charge in the oxide layer or/and at the interface. The expansion of this charge occurs mainly along the channel, while along the width of the channel the trapped charge is distributed homogeneously. Therefore effects of electrical stress is mainly 2D problem rather than 3D problem. Besides it 2D simulation is the considerably low time consuming simulation. Therefore we have preferred 2D simulation in our research.
- Compared to reality, there are simplifications. Can you please clarify them and evaluate their influence on results?
At line 65 it is included explanations of simplifications.
In reality the density of oxide trapped charge, induced by electrical stress, have some distribution along the channel, depending on stress condition [ ]. As our main purpose is to define whether has a place some law in the dependence of capacitance of the lateral junctions on linear size of the trapped charge, we considered only homogeneously charged area with average density of trapped charge. Indirectly to estimate how this simplification influence to results we considered the dependence of the threshold voltage change on linear size of homogeneously charged area (see chapter 3). The results is in good qualitative agreement with experimental results.
- Generally, the article has several serious flaws. The background theory is missing in the article. The article is based on simulations only. No experiments are proofing the presented results (I would like to say “presented ideas”; nevertheless, they need to be set-up on theory, which is missing).
The text about conformity of used physical model is added in line 71in section 3:
The effect of threshold voltage change is one of the main indicator of the electrical stress in MOSFETs. Mainly in experiments the dependence of threshold voltage change on the stress time. is measured. Besides it, it is well known that the increasing of stress time, under defined conditions, leads to expansion of the charged area. In purpose to check conformity the physical models used in the simulation to the real cases, the ΔVth dependence on the length of the oxide trapped charge is simulated. In the considered range of d the decrease of the threshold voltage, with an increase in the linear size d of the oxide charged area, is in qualitative agreement with the experimental results presented in literature.
The background theory is added in section 2 (line 48) and in section 3 (line 89)
- The word "measurement" is used at least three times in the article, including in the article title, which is confusing.
The word "measurement" used in the title, in the line 47 of the section 1 and the word "measuring" used in line 128 of section 3 means that the simulation results proposed in the paper can be used in the appropriate measurements.
7.Technical/formatting comments: the quantities and units in graphs are not matching the technical standards. Quality of the graphs (Fig. 2 and 3) is relatively low. The legend is missing in some figures; resp., it is useless (e.g. “B” in Fig. 1).
The Fig. 2 and Fig. 3 little bit is changed in accordance with changing the text. In Fig.1 the useless sign "B" is removed
Reviewer 2 Report
General comments
The authors are developing a simple modelling and simulation approach for the calculation of the effects of extension of the trapped positive charge in the gate oxide near the drain of the short channel MOSFET transistor on the threshold voltage and capacitances of the drain-substrate junction and source-substrate capacitance.
Overall, the manuscript can be published, but after the revision of the terminology and language used, and after improving the quality of the figures and associated legends. See below examples.
Specific comments
- The standard terminology for MOSFET transistor description is source-drain-substrate and associated junctions (drain-substrate and source-substrate). Instead of the transition region of drain-base, the term of depletion region of the drain-substrate junction should be used. The term of lateral transition should be replaced by lateral depletion region and so on.
- The legend of Fig. 1 should mention explicitly what was the size of the channel length which was used for those calculations. The term “linear size of charged area” does not mean anything. The authors should operate with length and width of the transistor channel. Therefore, the authors can specify the length of charge distribution along the channel length, towards the source region. The abbreviations like w.r.t. should be removed.
- Fig. 2 is not clear. In the text the authors speak about capacitances and on the vertical axis, the authors write variation of capacitance ΔC….
- The physical meaning of the increasing portion of capacitances as a function of voltage is not explained.
- Regarding the effect of trapped positive charge on the geometry of the depletion region near the interface gate oxide- p substrate, the authors need to discuss the level of positive trapped charge in the gate oxide which can create an inversion layer at the interface, as a major change in the geometry of the depletion region and its associated drain-substrate capacitance.
- A text coming from instruction to authors has remained in the manuscript, as shown below. Such text should be removed from the manuscript!
Authors should discuss the results and how they can be interpreted in perspective of previous 132 studies and of the working hypotheses. The findings and their implications should be discussed in 133 the broadest context possible. Future research directions may also be highlighted. 134
Author Response
Many thanks to reviewer for the valuable remarks, notes and suggestions, which helped to draw attention and I hope to improve the manuscript.
The responses and comments to the reviewer's remarks:
- The standard terminology for MOSFET transistor description is source-drain-substrate and associated junctions (drain-substrate and source-substrate). Instead of the transition region of drain-base, the term of depletion region of the drain-substrate junction should be used. The term of lateral transition should be replaced by lateral depletion region and so on.
In all parts of the paper the terms "drain-base transition" and "source-base transition" replased by "drain-substrate junction" and "source-substrate junction", "lateral transition" replaced by "lateral depletion region"
2.The legend of Fig. 1 should mention explicitly what was the size of the channel length which was used for those calculations. The term “linear size of charged area” does not mean anything. The authors should operate with length and width of the transistor channel. Therefore, the authors can specify the length of charge distribution along the channel length, towards the source region. The abbreviations like w.r.t. should be removed.
At line 64 in section 2 the term "linear size of charged area" is specified. The
abbreviation w.r.t. is removed.
3.Fig. 2 is not clear. In the text the authors speak about capacitances and on the vertical axis, the authors write variation of capacitance ΔC….
This is quite right remark. This is technical mistake, ΔCdb is replaced by Cds .
4.The physical meaning of the increasing portion of capacitances as a function of voltage is not explained.
In section 3 (line 89) the physical explanation of the increasing of capacitance at small voltages is added.
5.Regarding the effect of trapped positive charge on the geometry of the depletion region near the interface gate oxide- p substrate, the authors need to discuss the level of positive trapped charge in the gate oxide which can create an inversion layer at the interface, as a major change in the geometry of the depletion region and its associated drain-substrate capacitance.
The discussion about changing the geometry of depletion region and its association with changing the drain-substrate junction capacitance is added in the section 3.
6.A text coming from instruction to authors has remained in the manuscript, as shown below.
Such text should be removed from the manuscript!
The text coming from instruction to authors is removed.
Reviewer 3 Report
Report: Lateral C-V Measurement on Nanomosfet as a Method for Detecting the Hot Carrier Injection
General comments:
The paper is not particularly innovative and original as it describes a simulation a=of a well known process of threshold and capacitance voltage shift upon injection of external charge. The pararmeters described in Section 2, 100 nm gate length, gate oxide thickness of 3.7 nm and doping level of the p-Si substrate of 5E15 cm−3 are mentioned but not referred to any specific CMOS technology. Also the application field for this research is not mentioned at all. I also do not understand the two sections 4 and 5 for the conclusions, which in any case are quite poor. Also, the title refers to measurements while only simulations are described.
Author Response
Many thanks to reviewer for the valuable remarks, notes and suggestions, which helped to draw attention and I hope to improve the manuscript.
The responses and comments to the reviewer's remarks:
1.The paper is not particularly innovative and original as it describes a simulation a=of a well known process of threshold and capacitance voltage shift upon injection of external charge.
Yes, influence of injection of external charges to the parameters of MOSFET is well known
process. In the many papers it is considered the influence of injected charge to the threshold voltage, drain current even to capacitances (gate-to-drain, gate-to-substrate). However, the innovativeness of this paper is consideration the influence of trapped oxide charge to the lateral drain-substrate and source-substrate junctions.
2.The parameters described in Section 2, 100 nm gate length, gate oxide thickness of 3.7 nm and doping level of the p-Si substrate of 5E15 cm-3 are mentioned, but not referred to any specific CMOS technology.
The parameters of the simulated MOSFET was chosen from the range in which mainly electrical stresses was considered in the literature. This notation is added to the text in section 2 (line 65).
- Also the application field for this research is not mentioned at all.
The text about application field of the results is added in the conclusion (line 137)
4.I also do not understand the two sections 4 and 5 for the conclusions, which in any case are quite poor.
Introduction is added by the actual state-of-the-art and by declaring the authors’ way (advantages) compared to other authors’ work. All other sections considerably extended by including discussions and physical explanation of the carried out results.
5.Also, the title refers to measurements while only simulations are described.
The word "measurement" used in the title, in the line 47 of the section 1 and the word "measuring" used in line 128 of section 3 means that the simulation results proposed in the paper can be used in the appropriate measurements
Round 2
Reviewer 1 Report
Dear authors,
First of all, thank you very much for your effort.
I will follow the structure/remarks from my first review:
1, the recent content of abstract represents the article ideas.
2, the introduction is completed as well.
3, comments to 3D vs 2D model are explicit (perhaps a more detailed description of the model should be welcome; e.g. (based on Fig. 4) the boundary conditions and/or other data necessary for reproducing the same simulations).
4, here starts the critical parts of the review – the sentence: “The results are in good qualitative agreement with experimental results”, due to my opinion, must be followed by a numerical and/or verbal quantification. Nevertheless, the own experiments are still missing; the authors are following the experiments from Starkov and Starkov (cited as [28] in the article).
5, the background theory is deeper now; nevertheless, as mentioned above, the measurements are still missing. It is hard to review the simulations without having any comparison to measurements, or, at least, to results based on other simulation technologies.
6, please reconsider to replace the word “measurement” at least from the article title.
7, no more technical comments.
Author Response
Thank you very much for your effort and valuable review.
1, the recent content of abstract represents the article ideas.
Thank you very much.
2, the introduction is completed as well.
Thank you very much.
3, comments to 3D vs 2D model are explicit (perhaps a more detailed description of the model should be welcome; e.g. (based on Fig. 4) the boundary conditions and/or other data necessary for reproducing the same simulations).
In 2D simulation, the width of the device suitable for the third dimension in commercial software tools/programs as well as in the TCAD Sentaurus used in this work is usually 1 μm. Further, in many papers that take into account the effects of electrical stresses, the different values ​​of the widths are in a range that includes the value 1 μm (e.g. 0.47 μm and 0.87 μm in [27], 0.81 μm in [25], 10 μm in [30]). Let us mention, however, that the dependence on the width is beyond the scope of this work.
(The changes are introduced in lines 100-105)
4, here starts the critical parts of the review – the sentence: “The results are in good qualitative agreement with experimental results”, due to my opinion, must be followed by a numerical and/or verbal quantification. Nevertheless, the own experiments are still missing; the authors are following the experiments from Starkov and Starkov (cited as [28] in the article).
An evaluation/estimation and a quantitative comparison of the capacitance changes obtained from simulations and experiments were carried out. It can be seen (through simulation) that the maximum relative changes in the capacitance of the lateral drain-substrate are about 6%. Under certain conditions for introducing a charge into the oxide, this value lies in the range of variation of the lateral capacitance carried out in experiments (see Fig. 2 in [28], and Fig. 1 in [29]).
(The changes are introduced in lines 131-135)
5, the background theory is deeper now; nevertheless, as mentioned above, the measurements are still missing. It is hard to review the simulations without having any comparison to measurements, or, at least, to results based on other simulation technologies.
The experimental background for the method is provided by our previous works [28, 29]. These works show that the uneven (non-uniform) charge distribution in the oxide layer and at the oxide-semiconductor interface of the MOSFET leads to a change in the capacitances of the lateral drain-substrate and source-substrate junctions. Therefore the distribution of charges along the canal significantly affects the values of the aforementioned capacitances.
(The changes are introduced in lines 57-62)
6, please reconsider to replace the word “measurement” at least from the article title.
The title of manuscript “Lateral C-V Measurement on NanoMOSFET as a Method for Detecting the Hot Carrier Injection” was replaced by “Defining the Lateral C-V changes of NanoMOSFET as a Method for Detecting the Hot Carrier Injection”
7, no more technical comments.
Thank you very much.
Reviewer 2 Report
The authors have followed the recommendations and improved the manuscript. The manuscript can be published. For the future, a discussion of the impact of voltage threshold variation due to trapped charge on the transfer characteristics of the MOS transistor can be included as a convincing argument for this approach.
Author Response
Thank you very much for your effort and valuable review, recommendations and suggestions .
Reviewer 3 Report
Now the manuscript has been improved and my comments have been addressed
Author Response

(The authors gave the same response as above.)
